# Non-invasive monitoring of adrenocortical function in female domestic pigs using saliva and faeces as sample matrices

**Tanja E. Wolf** [1‡]*, **Norbert Mangwiro** [2‡]*, **Folorunso O. Fasina** [2,3], **Andre Ganswindt** [1,4]

**1** Mammal Research Institute, Faculty of Natural and Agricultural Sciences, University of Pretoria, Pretoria, Onderstepoort, South Africa, **2** Department of Production Animal Studies, Faculty of Veterinary Science, University of Pretoria, Pretoria, Onderstepoort, South Africa, **3** ECTAD, Food and Agriculture Organization of the United Nations, Tanzania, **4** Department of Anatomy and Physiology, Endocrine Research Laboratory, Faculty of Veterinary Science, University of Pretoria, Pretoria, Onderstepoort, South Africa

‡ TEW and NM share first authorship on this work.
* tanja.wolf@tuks.co.za (TEW); mangwiron@yahoo.com (NM)

**Data Availability Statement:** All relevant data files are available from the University of Pretoria Research Data Repository. https://researchdata.up. ac.za/articles/Non-invasive_monitoring_of_

## Abstract

Intensive pig management involves in a commercial setting the housing and implementation of certain procedures, such as castration and tail docking, which may be stressful for the animal. Good farming practices include the reduction of stress due to management processes, but assessing the level of stress perceived entirely through behavioural observations can be challenging. The monitoring stress-related physiological markers, like glucocorticoids (GC), can be an accurate alternative that would presumably be more objective. In order to avoid an additional stressor by taking blood, a non-invasive approach is advisable. We used an adrenocorticotropic hormone (ACTH) stimulation test and the effect of transport to examine the suitability of different enzyme immunoassays (EIAs) for monitoring adrenocortical function in domestic pigs using saliva and faeces as sample matrices. An assay measuring faecal glucocorticoid metabolites (fGCMs) with a 3ß,11ß-diol group has proven suited to determine adrenocortical activity, showing an overall increase of 180% in fGCM concentrations related to ACTH administration and of 70% related to transport, respectively. A cortisol EIA was used to detect salivary glucocorticoid (sGC) concentrations, revealing a 1100% increase in sGC concentrations after ACTH administration. The stability of fGCM concentrations post-defecation was determined to assess possible changes in measured fGCM concentrations in unpreserved faecal material over time, with fGCM concentrations being relatively stable (maximal 12% change) under natural conditions for approximately two days after defecation. This implicates that untreated faecal material from pigs can be analysed for up to two days after collection without appreciable level of depreciation in fGCM concentrations. Being able to assess the physiological stress response of domestic pigs non-invasively can help to improve the well-being of commercially reared pigs.

adrenocortical_function_in_female_domestic_
pigs_using_saliva_and_faeces_as_hormone_
sample_matrices/11763420 (DOI: https://doi.org/
10.25403/UPresearchdata.11763420.v1).

**Funding:** Funding for this work was provided partly by the National Research Foundation Incentive funding for Rated Researchers (2016) provided to Fasina FO. The funders had no role in study design, data collection and analysis, decision to publish, or preparation of the manuscript.

**Competing interests:** The authors have declared that no competing interests exist.

## Introduction

Over 1 billion pigs (*Sus scrofa domesticus*) can be found worldwide [1], and a large proportion of these animals are intensively managed in confined pens and sow crates, and these housing conditions have raised questions and concerns on animal welfare [2–4]. In this regard, animal welfare issues are often evaluated by monitoring proxies for perceived intrinsic and extrinsic stress [5]. Potential stressors could be periods of weaning, transportation, mixing, and confinement, as well as events like castration and tail docking among others [6]. As pigs are very susceptible to stressors, those can potentially influence the quality of the meat [7]. Due to animal welfare concerns and rising consumer interests, the management and housing conditions of pigs and its potential stress-inducing effects are of utmost interest to the industry [4,8]. So far, it is common practice within the pig farming industry to evaluate stress caused by management practices using indicators like vocalization or stereotypic behaviours, which can be susceptible to observer bias [8]. An alternative approach would be the monitoring of stress-related physiological markers, like glucocorticoids, which would presumably be more objective [3].

In response to a perceived stressor, corticotropin-releasing hormone (CRH), released from the hypothalamus will stimulate the pituitary to release adrenocorticotropic hormone (ACTH), which in turn stimulates the adrenal cortex to release glucocorticoids [9,10]. Glucocorticoids (GC) can therefore be seen as key elements in the neuroendocrine stress axis, and their measurement can give insight into an animal's well-being [11]. So far, blood, saliva, and to a quite lesser extent urine have been used as hormone matrix for monitoring glucocorticoid alterations in pigs [12–15]. Although blood is still wildly used for determining glucocorticoid concentrations in a wide range of species including pigs, respective results are often of limited value as determined steroid levels are usually affected by episodic fluctuations and a diurnal pattern of hormone secretion [16]. In addition, blood collection often requires restraining of the animal, which is usually perceived as a stressful event in itself [17]. Consequently, minimal- or even non-invasive monitoring of steroid hormone metabolites, using alternative matrices like saliva, urine, and especially faeces, have become an increasingly popular technique [18,19]. However, the collection of saliva still requires some manipulation of the animal, and thus cannot be regarded as entirely feedback free. Furthermore, it can only be applied to a limited extent in free-moving animals [16,20]. Finally, salivary steroid concentrations are also susceptible to circadian rhythm and episodic fluctuations of hormone secretion, and are therefore not ideal, as sampling needs to take place at a standardized time of the day when investigating chronic stress. Using faeces as basic sample material for determining glucocorticoid output in pigs would be an ideal approach, as it does not involve handling and therefore can avoid alteration of the results due to handling stress. Furthermore, faecal samples are less affected by daily variation in hormone secretion [21], and can be comparatively easily collected even under free-roaming conditions. So far, however, attempts to reliably determine adrenocortical activity as a measure of stress in domestic pigs using faeces as sample matrix have led to inconclusive results, leading to doubts whether glucocorticoid alterations can actually be measured reliably using pig faeces [22].

Therefore, the overall aim of this study was to examine the suitability of four enzyme immunoassays (EIAs) for monitoring adrenocortical function in domestic pigs using saliva and faeces as sample matrices. More specifically, the objectives of this study were to: a) determine stress-related physiological responses in domestic pig's saliva and faeces by performing an adrenocorticotropic hormone stimulation test (ACTH challenge test), b) determine the effect of circadian rhythm on salivary glucocorticoid (sGC) and faecal glucocorticoid metabolite

(fGCM) levels, c) verify the effect of transport on fGCM levels, and d) investigate the stability of fGCM levels post-defecation.

## Material and methods

### Study animals and housing

Six Large White-Landrace F1 gilts (weight between 43.7 and 47.7 kg), approximately 14 weeks of age, were housed at the Biomedical Research Centre (UPBRC), University of Pretoria, South Africa, for the study. After the initial transport to the UPBRC, the pigs were allocated to a straw-bedded pen for acclimatisation to the new environment for 3 days. Thereafter, the animals were individually-housed in metabolic crates and had *ad libitum* access to food and water. The housing conditions (e.g. temperature, humidity) were monitored and the crates were cleaned on a daily basis.

After the end of the experiment, the animals were moved to standard grow-out pens and not used any further. The entire study was performed under the approval and in accordance with the guidelines of the Animal Ethics Committee of the University of Pretoria (V094-13).

### Monitoring of transport and ACTH challenge test

The duration of the entire study was 17 days. To investigate the effect of relocation to biologically validate the most suited EIA, one faecal sample per day was collected from each animal from the day of arrival at UPBRC for three consecutive days (n = 18). No sampling took place for the following four days of acclimatisation. For determining individual baseline values faeces were again collected for 6 days prior ACTH injection. On the day of ACTH administration (early morning), the pigs were minimally restrained by blocking each animal off to a side of the metabolic crate without the use of force. Four of the six animals (T1 –T4) were each injected with approximately 50 IU (10 µg/kg) of ACTH analogue (Synacthen© Depot, Norvartis Switzerland) in the neck muscle. The remaining two animals served as controls (C1 + C2) and received 0.5 ml physiologic saline (0.9% Sodium chloride) each through the same route. Sample collection continued on day of injection and for another 3 days post injection. In addition, saliva samples were collected frequently from three days prior to three days post ACTH injection.

### Sample collection

Throughout the study, saliva was collected, at least twice a day, once between 08:00–08:30 h in the morning and again between 15:00 and 15:30 h in the afternoon. On the day of ACTH/ Saline injection, salivary samples were collected approximately an hour prior administration and subsequently 20, 40, 60, 90, and 120 min post injection. Thereafter, saliva was collected at hourly intervals until 8 hours post-administration, and further twice daily until the end of the experiment. Saliva was collected using Salivette® tubes (Sarstedt, Germany). The sponges were either held using forceps or hanged into the crate with a transparent string in order to allow the pigs to chew on it for 1 to 2 min.

To investigate the effect of transport on fGCM levels, individual faecal samples were collected once per day between 12:00 and 18:00h. After the four days of full acclimatisation, faecal samples were collected daily between 08:00 and 18:00h starting six days before the ACTH injection until three days post- injection. During this period, samples from all voided faeces were collected. Faecal samples were collected as soon as they were voided using the outlets on the base and the sides of the metabolic crates. Using gloves, about 10 to 15 g of freshly voided

material was taken from the core of each faecal sample, mixed and placed in a sterile plastic container.

Immediately after collection, faecal and salivary samples were stored at -20˚C and kept frozen until further processing. A total of 161 faecal samples (n = 72 prior and n = 89 post injection) and 170 saliva samples (n = 42 prior and n = 128 post injection) were collected throughout the experiment.

### Effect of collection time and degradation study

In order to investigate a possible influence of collection time on sGC as well as fGCM concentrations, steroid concentrations determined in salivary (six days of collection) and faecal (six days of collection) samples collected in the morning were compared to respective steroid values from samples collected in the afternoon.

To determine the effect of increasing time intervals between sample defecation and freezing on the stability of fGCM concentrations, freshly voided faeces from three pigs were collected additionally, prior to injection. The faecal material was homogenized, and divided into 24 equal sub-samples, which were then stored at ambient room temperature (22˚C to 28˚C). At different time points (0, 1, 2, 4, 8, 16, 24, and 48 hours post-defecation) three sub-samples were taken and stored frozen at -20˚C until analysis [23].

### Steroid extraction

The saliva was obtained from the sponge following the protocol by Ruis and colleagues [6]. In brief, the thawed sponge was centrifuged for 10 min at 1500g. The retrieved saliva was then stored at -20˚C until further analysis.

The frozen faecal material was lyophilized, pulverized, and sieved through a mesh to remove remaining fibrous material [24]. For steroid extraction, 0.100 to 0.110 g of the faecal powder was vortexed for 15 minutes with 3 ml of 80% ethanol. After centrifugation at 1500g the supernatants were transferred into microcentrifuge tubes and stored at -20˚C until analysis [23].

### Steroid analysis

Retrieved saliva was measured for salivary glucocorticoid (sGC) concentrations via a cortisol enzyme immunoassay (EIA) using an antibody against cortisol-3-CMO:BSA [25]. Further details on the assay characteristics including a full description of the assay components and antibody cross-reactivities are described in Palme & Möstl 1997 [25]. The sensitivity of the EIA was 0.6 ng/ml, and the intra- and inter-assay coefficients of variation (CVs), determined by repeated measurements of high and low value quality controls, were 9.5% and 11.0% (for Intra-CV) as well as 15.0% and 15.2% (for Inter-CV), respectively.

To identify a suitable EIA for effective assessment of adrenocortical function using faeces as sample matrix, a selected set of faecal extracts (n = 31) from three pigs were measured for faecal glucocorticoid metabolite (fGCM) concentrations. Four different enzyme immunoassays were tested, namely (i) a 11-oxoaetiocholanolone I EIA (detecting 11,17 dioxoandrostanes; 11,17-DOA), (ii) 11-oxoaetiocholanolone II EIA (detecting fGCMs with a 5β-3α-ol-11-one structure; 3α,11-oxo-CM), (iii) a 5α-pregnane-3β,11β,21-triol-20-one EIA (measuring 3β,11β-diol-CM), and (iv) the above mentioned cortisol EIA. Detailed assay characteristics, including full descriptions of the assay components and cross-reactivities were previously described by Palme and Möstl [25] for the 11,17-DOA and cortisol, by Möstl et al. [26] for the 3α,11-oxo-CM, and by Touma et al. [27] for the 3β,11β-diol-CM EIA. After identifying the most suitable EIA with regards to fGCM elevation post-ACTH administration, the entire sample set,

including the material from the stability experiment, was assessed using only the 5α-pregnane-3β,11β,21-triol-20-one EIA.

The sensitivity of the assays were 0.6 ng/g faecal dry weight (DW) for 11, 17-DOA, 3α, 11-oxo-CM and cortisol, and 1.5 ng/g DW for the 5α-pregnane-3β, 11β,21-triol-20-one EIA. Intra-assay coefficient of variations, determined by repeated measurements of high and low value quality controls, were 9.5% and 11.0% for the cortisol EIA, 4.9% and 5.9% for the 11, 17-DOA EIA, 6.1% and 8.0% for 3α, 11-oxo-CM EIA, and 4.9% and 6.3% for the 5α-pregnane-3β, 11β, 21-triol-20-one EIA. Inter-assay coefficient of variation, only determined for the 5α-pregnane-3β, 11β, 21-triol-20-one EIA by repeated measurements of high value quality controls, was 9.5%. All assays were performed on microtiter plates following established protocols [28] at the Endocrine Research Laboratory at the Faculty of Veterinary Science, University of Pretoria.

## Data analysis

To determine reliability of an EIA by showing an adequate increase in glucocorticoid output post ACTH-administration samples from three individuals from 48 hours pre-injection until 36 hours post injection were used. To establish a baseline individual median fGCM concentrations were calculated using pre- ACTH administration samples. Peak fGCM concentrations following ACTH administration were expressed as percentage increase above baseline concentrations.

Subsequently, determined sGC and fGCM concentrations from all individuals were compared to respective individual baseline levels using the chosen assays. Descriptive statistics were used to describe peak increase after ACTH/saline injection in percentage for sGC and fGCM concentrations.

To biologically validate the chosen EIA the initial transport of the animals was used and the fGCM concentrations in percentage were compared to the subsequent individual baseline levels during the acclimatisation period.

To evaluate the effect of time of day on GC concentrations, samples collected in the morning (saliva: 8:00–8:30h, faeces: 8:00–10:00h) were compared to samples collected in the afternoon (saliva: 15:00–15:30h, faeces: 12:45–18:00h). On six different days, a minimum of one sample per matrix per individual for all six study animals was collected in the morning and afternoon, respectively (total n = 2x 36 samples per matrix) In case that more than one sample was collected from an animal within a respective time window on a given day the mean steroid concentration of those samples was used for analysis. Individual sGC, as well as fGCM concentrations from morning samples were compared to individual afternoon samples using a Wilcoxon signed rank test.

The relative fGCM metabolism rate post defaecation (%) was calculated on pooled samples using the mean value determined at t = 0 as 100%. Differences in fGCM concentrations between sampling subsets were subsequently described as change in percentage. All statistical analyses were performed using R, version 3.0.2 [29].

## Results

### ACTH challenge test

Administration of synthetic ACTH resulted in an overall median increase of 1100% above baseline in sGC levels (Fig 1), with peak concentrations occurring 40 to 90 minutes after injection. Administration of saline (controls) was reflected in a median increase of 400% above baseline sGC concentrations with peak concentration also present 40 to 90 minutes after

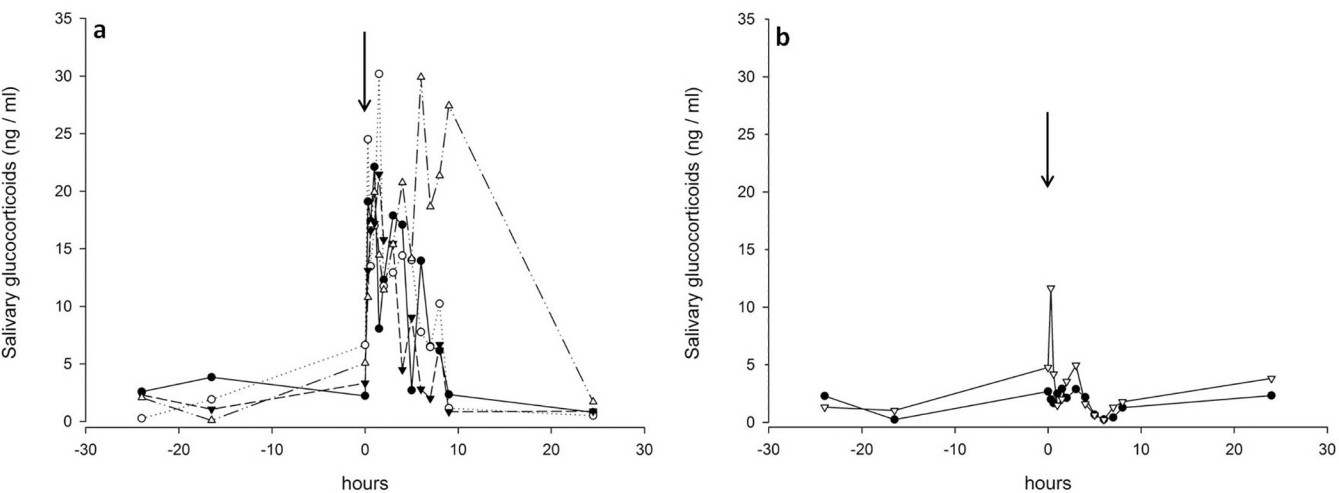

**Fig 1.** Salivary glucocorticoid (sGC) concentrations of six pigs 24 hours prior- and post injection of synthetic ACTH (a) or saline (b). Arrow indicates time of injection.

injection. Overall sGC concentrations returned to baseline 5 to 24 h post injection for ACTH, and 1 to 4 h for the controls, respectively.

All four assays tested for fGCM concentrations revealed an overall increase between 110% and 180%, and peak values occurred within 12 to 24 h after injection (Table 1). However, the 5α-pregnane-3β, 11β,21-triol-20-one EIA showed the most comparable baseline and peak values for all three animals and was chosen for the analysis of the remaining faecal samples.

Overall baseline fGCM level, using the 5α-pregnane-3β,11β,21-triol-20-one EIA, for all 6 pigs was 226.4 ng/g dry weight (DW) (range 215.7 to 255.1 ng/g DW). Administration of ACTH resulted in an overall 180% (range: 60–220%) increase above baseline in fGCM levels, with peak concentrations present 24 hours after injection (Fig 2). Overall fGCM concentrations returned to baseline 36 to 48 hours post injection.

## Effect of transport on fGCM

The initial transport of the pigs to the experimental facility was used as a biological validation of the chosen EIA. The transport was reflected in overall increased alterations of fGCM levels of 70% against subsequent baseline values between 24 and 48 h after the event (Fig 2). The

**Table 1. Baseline and peak fGCM concentrations, and overall median increase (%) for samples from the three animals tested after ACTH injection with four enzyme immunoassays (EIAs).**

| EIA | 11-oxoaetio-cholanolone I | 11-oxoaetio-cholanolone II | 5α-pregnane-3β,11β,21-triol-20-one | Cortisol |
|---|---|---|---|---|
| Individual median fGCM baseline (ng/g DW) | 31.0 | 127.4 | 234.3 | 35.4 |
| | 57.8 | 273.9 | 224.7 | 51.1 |
| | 97.9 | 272.1 | 205.7 | 25.3 |
| Individual peak fGCM levels post ACTH injection (ng/g DW) | 44.9 | 289.3 | 419.5 | 61.8 |
| | 162.0 | 765.1 | 549.0 | 105.8 |
| | 372.5 | 1373.7 | 559.8 | 68.6 |
| Overall median increase | 180% | 180% | 140% | 110% |

DW = dry weight.

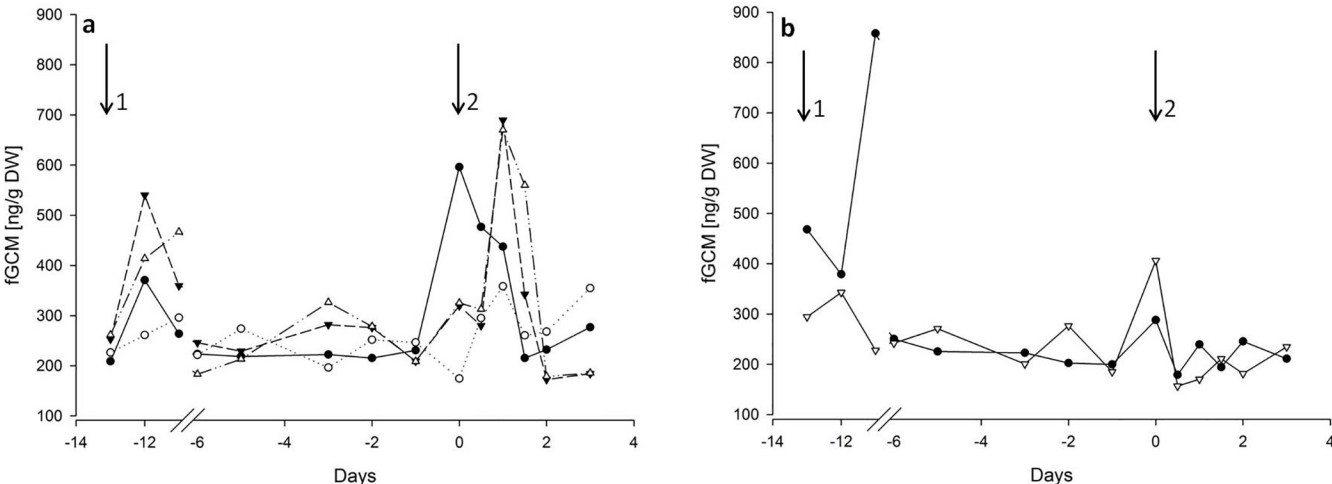

**Fig 2.** Longitudinal profiles of faecal glucocorticoid metabolite (fGCM) concentrations of six pigs monitored directly after arrival, adaptation period, prior- and post injection of synthetic ACTH (a) or saline (b) using a 5α-pregnane-3β,11β,21-triol-20-one EIA. Arrows indicate transport (1) and injection (2).

fGCM levels decreased during the acclimatisation period and reached baseline levels seven days after arrival at the experimental facility.

## Effect of collection time

A significant difference was found in sGC concentrations when comparing samples collected in the morning compared to samples collected in the afternoon (V = 571, p < 0.001), with sGC concentrations collected between 08:00–08:30h being higher than sGC concentrations collected between 15:00 to 15:30h. In contrast, no significant difference was found, when comparing fGCM concentrations of samples collected either in the morning (08:00–10:00h) or the afternoon (12:45–18:00h, V = 315, p = 0.786).

## Stability test for fGCM concentrations

Within the first hour at ambient temperature, fGCM concentrations showed an initial decrease of 12%. The fGCM concentration than stabilized over the next 24 hours at approximately 90% of the initial concentration (Fig 3). Variation in fGCM concentration between sample subsets ranged from 2.3 to 15.2% for respective measuring points in time.

## Discussion

Reliable monitoring of adrenocortical activity in a particular species using fGCMs requires a solid validation of the test system in question as different animal's gastrointestinal transit (GIT) systems and microbiome activities may differentially produce different fGCM [9,15]. The main route of excretion for glucocorticoids in pigs is urine and so far no assay system has been proven suited for measuring fGCM in pigs [20]. Previous studies have used an ACTH challenge, as well as a dexamethasone suppression test [22], or a biological validation [30] to identify a suitable test system using faeces as sample matrix, however with mixed results. After the injection of ACTH in only three out of six animals an increase in glucocorticoid metabolites could be measured using an 11-oxoaetiocholanolone I EIA [22], while the biological validation comparing pre- and postweaning samples confirmed the validity of the used corticosterone EIA [30]. In this regard, our study demonstrates a non-invasive technique to

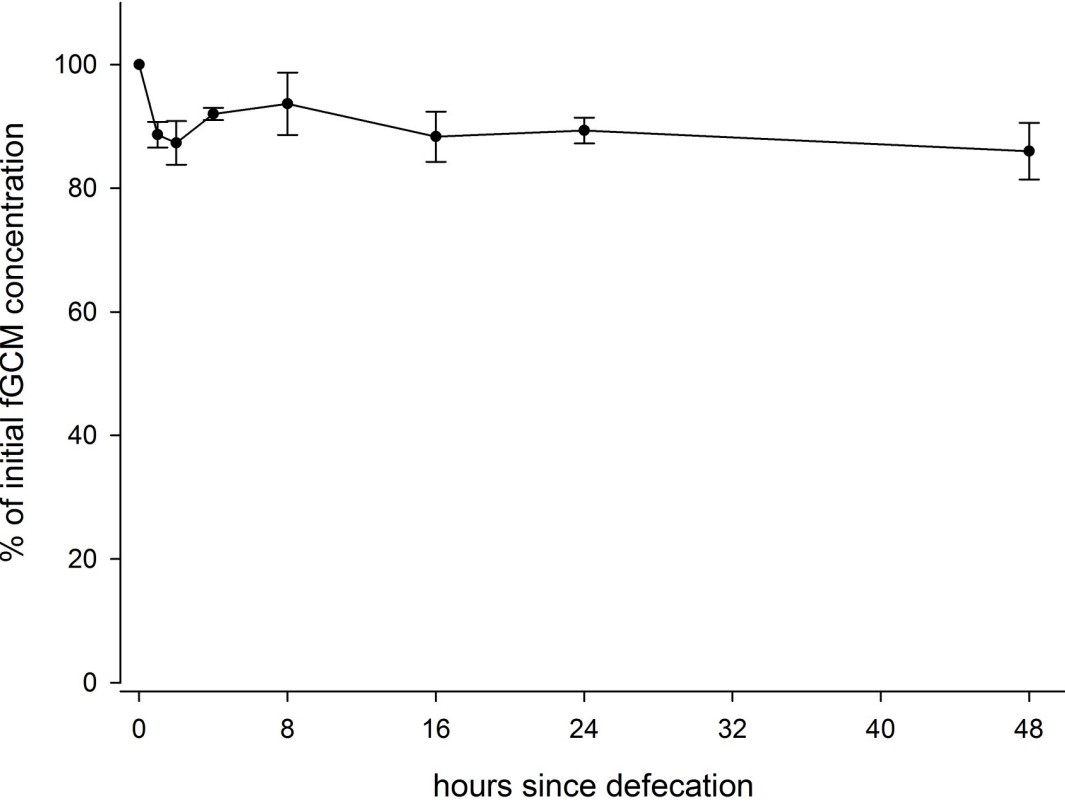

**Fig 3. Relative change (%) of faecal glucocorticoid metabolite concentrations (mean +/- SE) in pigs over time since defecation (0, 1, 2, 4, 8, 16, 24, and 48h).**

monitor adrenocortical function based on measuring the concentrations of glucocorticoid metabolites in faeces, using an ACTH challenge test for physiological validation and using the initial transport event as an additional biological validation.

For the saliva samples, a cortisol assay (using an antibody against cortisol-3-CMO:BSA) has been shown to reliably detect salivary GC concentrations. An increase in sGC levels of over 1000% occurred within the first 90 minutes post injection, which is consistent with the findings of Bushong et al. 2000 [31], who also observed a four to six fold peak response of sGC in barrows. The results found in pigs are comparable with other species. In horses for example a significant increase in sGC was found 30 minutes post injection, with a peak after approximately 2 hours [32]. In ewes a similar timeframe was found with a peak in salivary cortisol release approximately 60 minutes after ACTH administration [33].

After the initial screening of 4 available enzyme immunoassays (EIAs) for the measurement of faecal glucocorticoid metabolites an assay using an antibody against 3β,11β-diol-CM has been identified to suitably determine fGCM concentrations in female domestic pigs. Peak fGCM concentrations post injection showed an increase of approximately 180% compared with the baseline values about 24 hours post injection before returning back to baseline values 48 hours post injection. Although individual baseline levels varied between 215.7 and 255.10 ng/g DW, all individuals reacted to the ACTH/saline injection with increased fGCM levels. In horses, a similar peak between 200 and 600% in fGCM levels could be observed between 16 and 34 hours post ACTH injection. In contrast to our study the values in the horses decreased

back to basal concentrations after already half a day [22]. In goats a similar long period of elevated fGCM could be found, here the levels did not reach baseline after 24 hours [34].

Whereas sGC concentrations rapidly returned to baseline following administration of ACTH, a delay in fGCM concentrations could be seen and this is most likely attributed to the intestinal passage, which causes a time delay between the circulation of steroids in the plasma and the appearance of their metabolites in the faeces [35].

Not only the actual transport can be a stressful event for animals, but also being in a transporter itself can lead to an increased glucocorticoid release. In pigs it has been shown, that blood cortisol levels increased significantly after being loaded onto a transport vehicle [36]. When the vehicle was moving the plasma cortisol levels remained elevated for a longer time period compared to the vehicle staying stationary. Although the transport in our study caused an increase in fGCM levels, this increase is distinctively lower than the increases found in other studies, for example in sheep an increase of approximately 300% was observed [37]. The observed differences may be due to the durations and conditions of transportations, as well as due to species and measurement differences. The pigs in this experiment were transported for about 20 minutes on a smooth road whereas in the experiment of Lexen et al. 2008 [37], the animals were transported for approximately 60 minutes.

As with many other hormones, plasma glucocorticoids have a circadian rhythm in most vertebrate species [16,20], and diurnal variations of fGCMs were already demonstrated for at least some mammals [16]. In pigs, the basal concentrations of cortisol in blood are generally higher in the morning than in the afternoon and evening [6]. Our findings showed that there was a significant difference in sGC concentrations in samples collected in the morning compared to those collected in the afternoon, with higher concentrations in morning samples. This has also been observed in other studies comparing morning and evening samples [6,38,39]. The results confirm that circadian rhythmicity of cortisol should be taken into consideration when evaluating stressors in pigs using saliva as a sample matrix. However, fGCM concentrations did not show a significant difference between morning and afternoon samples and thus might be a more suited approach to monitor adrenocortical activity if frequent sampling at a specific time cannot be guaranteed.

The stability of fGCMs post-defecation has been determined to assess possible changes in measured hormone concentrations in unpreserved faecal material. Steroid concentrations varied by 13% over a period of 48h. These changes agree with other studies, that showed an increase in fGCM levels within a few hours (4–24 h) post defecation (cattle & horses [22]; sheep [37]). However, a rather negligible change in fGCM levels over several days has been reported in other species [40–42]. While results can differ distinctively even when using the same assay but in different species and vice versa [9,37], to date, there is no empirically conclusive explanation for this observation other than bacterial enzymes and microbiome activities. In different animal species different bacteria may be present in the faeces which further metabolize fGCMs differently [16,43]. Therefore, it is necessary to test storage methods if immediate freezing of samples is not possible [9].

From the findings in this study, fGCM in pig faeces was relatively stable for approximately two days. The implication of this finding is that faecal materials from pigs can be collected from distant locations and transported to the laboratory for up to two days without appreciable level of depreciation in fGCM concentrations.

Potential limitations of the current study is the sex and the age of the animals with only immature females. Further studies with mature individuals of both sexes would be helpful to elucidate the suitability of the described assay.

## Conclusion

Animal well-being can be defined as the animal living in harmony within the environment, meaning that it is able to adapt to it physically as well as psychologically [44]. Measuring psychological well-being of an animal is difficult, and therefore evaluating health and behaviour are an alternative [45]. Being able to non-invasively assess the physiological stress response of domestic pigs by using faeces as a sample matrix can help to improve the well-being of commercially reared pigs. Potential applications could for instance be the possibility of measuring the potential effect of different rearing practices on stress in pig, or more specifically the ability to measure potential variability of fGCM levels in different pig housing set ups, such as group housing or confined single stands.

Therefore, the generated information regarding pigs should help to facilitate further studies which examine endocrine responses to putative stressful circumstances in different pig environments and set ups.

## Acknowledgments

The authors would like to thank S. Ganswindt and H. Roussow for their expert help with laboratory techniques.

## Author Contributions

**Formal analysis:** Tanja E. Wolf, Norbert Mangwiro.

**Investigation:** Norbert Mangwiro.

**Methodology:** Folorunso O. Fasina, Andre Ganswindt.

**Resources:** Folorunso O. Fasina, Andre Ganswindt.

**Supervision:** Folorunso O. Fasina, Andre Ganswindt.

**Writing – original draft:** Tanja E. Wolf, Norbert Mangwiro.

**Writing – review & editing:** Tanja E. Wolf, Folorunso O. Fasina, Andre Ganswindt.

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
