## [Decision Letter · Decision Letter 0]

23 Dec 2019

PONE-D-19-24903

Non-invasive monitoring of adrenocortical function in domestic pigs using saliva and faeces as hormone matrices

PLOS ONE

Dear Dr Wolf,

Thank you for submitting your manuscript to PLOS ONE. After careful consideration, we feel that it has merit but does not fully meet PLOS ONE’s publication criteria as it currently stands. Therefore, we invite you to submit a revised version of the manuscript that addresses the points raised during the review process.

As suggested by Reviewer 1, the manuscript needs to be revised thoroughly before it can be considered for acceptance.

We would appreciate receiving your revised manuscript by Feb 04 2020 11:59PM. To enhance the reproducibility of your results, we recommend that if applicable you deposit your laboratory protocols in protocols.io, where a protocol can be assigned its own identifier (DOI) such that it can be cited independently in the future. For instructions see: http://journals.plos.org/plosone/s/submission-guidelines#loc-laboratory-protocols

We look forward to receiving your revised manuscript.

Kind regards,

Govindhaswamy Umapathy, PhD

Academic Editor

PLOS ONE

Journal Requirements:

2. In your Methods section, please include a comment about the state of the animals following this research. Were they euthanized or housed for use in further research? If any animals were sacrificed by the authors, please include the method of euthanasia and describe any efforts that were undertaken to reduce animal suffering

Reviewers' comments:

Reviewer's Responses to Questions

**Comments to the Author**

1. Is the manuscript technically sound, and do the data support the conclusions?

Reviewer #1: Partly

Reviewer #2: Yes

2. Has the statistical analysis been performed appropriately and rigorously? 

Reviewer #1: Yes

Reviewer #2: Yes

3. Have the authors made all data underlying the findings in their manuscript fully available?

Reviewer #1: No

Reviewer #2: Yes

4. Is the manuscript presented in an intelligible fashion and written in standard English?

Reviewer #1: Yes

Reviewer #2: Yes

5. Review Comments to the Author

Reviewer #1: The paper of Wolf et al (PONE-D-19-24903) aims at validating a method for evaluating adrenocortical activity via measuring faecal glucocorticoid metabolites (fGCMs) in pigs. The authors performed an ACTH challenge test (physiological validation) and a biological validation (transportation) in 6 female, immature pigs. They tested 4 different EIAs on a subset of sample (3 pigs) and found a 5α-pregnane-3β,11β,21-triol-20-one EIA best suited. Clear increases after ACTH injection and transportation in all 6 animals successfully validated the chosen EIA. An additional storage experiment demonstrated that FGCMs were stable up to 48 h. I think the experimental setup is well designed and the study adds knowledge about fGCMs analysis in an important species, where such methods have not been successfully validated so far. However, I miss a clear statement about the limitations/weaknesses of the study. The authors only used female, immature pigs. Because reproductive status (age) and sex are serious confounders in fGCM analyses, their influence needs to be elucidated (and the method validated in mature animals of both sexes) before a broad application of this method in pigs can be recommended.

Detailed comments (ordered by appearance in the ms).

Title: I suggest indicating that only females (in female domestic pigs, or sows) were used (important as sex differences might be present). Better use “sample matrices” instead of “hormone matrices” (only metabolites are present in faeces)

Abstract: line 24: Replace “In order of not adding” with “To avoid”.

Line28: “sample matrices” (also line 80, 150)

I suggest to reword: “An assay measuring faecal glucocorticoid metabolites (fGCM) with a 3ß,11ß-diol group has proven suited to determine adrenocortical activity , showing…” (the antibody is not against those metabolites).

Line 31: Better: A cortisol EIA was used… (the immunogen is cortisol-3-CMO:BSA)

Lines 33-38: That could be described shorter (especially as collection and transport from distant locations might be a problem in wildlife, but unlikely in domestic pigs, or?)

Line 52: stereotypic

Line 67: Your focus is domestic livestock – so would not be another review better suited here?

Somewhere in the introduction, it might be worth mentioning that pigs are an especially stress susceptible species.

Line 85: ..on the stability of….

Line 90: I wonder when females of this breed get sexually mature, but 14 weeks old animals are too young.

Line 94: normally written in italics and with 2 words: ad libitum

Line 103: 50 IU (10 µg) there is a space between a number and its dimension (many other cases too – e.g. line 134; 138, etc..).

Line 104: Add country! Switzerland?

Line 106: Were samples from all defecations collected? Here and elsewhere: I suggest to replace the hyphen with a word space.

Line 114: Salivette, or?

Line 117-118: I guess the portion was taken from the core of each samples, and then mixed? Or why is it important to take it from the core of a well-mixed sample?

Line 125: .. between defecation and freezing, or? For stability it is less important when the sample was collected!

Line 128-129: I can imagine what was performed, but wording is unclear.

Line 142: via a cortisol enzyme immunoassay…. against cortisol-3-CMO:BSA.

Line 149: “Suited” is better than “reliable”. How would you define the latter? Most important is an EIA, which can monitor expected increases/decreases in adrenocortical activity well.

Line 151 onwards: Four different EIAs were tested – please add EIA to each of the descriptions (e.g. 11-oxoaetiocholanolone I EIA) - I guess GB-spelling is chosen, so aetiocholanolone is correct.

Lines 156-157: not possible to write by [number] – need to write the authors and then [].

Line 159: delete “fGCM”

Line 175-176: Peak fGCM concentrations following ACTH administration were expressed as percentage increase above baseline concentrations.

Line 183-186: How many samples per animal were present (I guess only baseline samples were used)?

Line 187: post defecation.

Line 188: mean value … (no hormones in the faeces).

Line 188/189: Unclear what the authors mean here? Why distribution?

Line 195: ..occuring

Line 209: enzyme immunoassays (EIAs)

Table 1: The two 11-oxoaetiocholanolone have a higher increase – so why was the 5α-pregnane-3β,11β,21-triol-20-one EIA chosen? What means a “more stable” baseline? How is this made objective?

Looking at Möstl et al (1999) seems to show that the 11-oxoaetiocholaonolone I had even higher increases – but it proved unsuited, as not all animals showed the increase, and dexamethasone injection failed to clearly suppress levels. Most important: fGCM were not stable after defecation (a 175% increase after 4 h, 375% after 24 h). The latter may be an argument for choosing the 5α-pregnane-3β,11β,21-triol-20-one EIA.

Line 213: (range: 215.7…) – but the 205.7 given in Table 1 is lower, why that?

Fig 2 (and the same for Fig. 1): How were those 4 chosen (of course this makes someone curious about the other 2 ones (especially as Möstl et al., 1999 found even higher increases in some pigs, but no expressed ones in others). Please show all 6 pigs.

Line 232 (235) – what does “V” mean?

Line 248: first time of “GIT” – spell it out!

Line 249: add [8] to [14]

Line 250: “suited” is better than “sufficient”

Line 251: [20] they also used a dexamethasone suppression test, and the assay was found unsuited, based (at least partly) on the fact that the suppression was not well reflected in fGCM levels.

Line 250/251: Those studies did not perform ACTH test or biological validations “to measure fGCM concentrations” – please reword.

Line 256: .. a cortisol assay (using an antibody against cortisol-3-CMO:BSA) …

Line 259: barrows? Did you mean “farrows”?

Line 263: no hyphen in case of enzyme immunoassay. “glucocorticoid metabolites” – and also reword the rest (see comments above).

Line 265-267: Please reword. I wonder, but cannot find a sustained (up to 48 h) peak in Fig 2?

Line 268: baseline levels between 221.5 and 246.3 ng/g: I’m confused, as this is the third interval given for the baseline levels (Tab 1; line 213), and all are different?

Line 269 – repetition of line 266 (180% increase)?

Line 270: delete one full stop.

Line 272: .. and the levels did not reach baseline after 24 hours [32].

Line 275: “fGCM recovery time”?

Line 276: .. and the appearance of their metabolites in the faeces [33].

Lines 283-286: I suggest adding “other species” and “different measurement method” as the most likely reasons for the observed differences.

LINE 288: “were” instead of “was”

Line 294: .. when evaluating stressors in pigs using saliva as sample matrix.

Line 296: ..might be a more suited..

Line 299 “steroid concentrations”

Lines 298-315: this is a rather long passage dealing with the stability experiment. I suggest focusing on domestic livestock and shortening it (e.g. the sentences lines 307-310 can be easily deleted without losing much). As mentioned before, I doubt there is much application in pigs for collection in remote locations and long sample transport.

Line 322 onwards: I would prefer more straight forward applications in immature pigs (too many potential things/possibilities) – but see also further comments below!

Lines 326-327: Don’t think that this statement is true. Your aim was validating such a non-invasive tool, but the importance must be beyond the tool.

A clear statement about the limitations/weaknesses of the study needs to be added here. Only female, immature pigs were used. Thus results don’t guarantee a broad application of the method in pigs, because reproductive status (age) and sex are serious confounders in fGCM analyses, and thus their influence needs to be elucidated. That clearly limits the application of the method!

References: Not all words should be capitalized in the title [correct for 6, 9, 12,15, 17, 22, 28, 33, 36, 46].

Line 337: last accessed?

Line 346: delete the 1 after cattle.

Line 379: Domestic Animal Endocrinology

Line 388: the journal is missing.

Line 393: “Suppl 2” is missing.

Line 395: Veterinary Research Communications

Line 452: “ethological”

Line 453: better: “last accessed” than “updated”.

Fig 1 and 2: Smaller symbols for the 4 pigs would make the picture clearer, also allowing adding the missing two.

Fig. 2. What about the animal which showed a peak in fGCM concentrations at the time of ACTH injection? Was there a known stressor occurring the day before? There seemed to be a smaller peak (ACTH related) present on Day 1 (based on the broader shoulder).

Fig. 3: The y-axis should be scaled from 0 to 100. This is more correct and also better representing the message of only insignificant changes in FGCM levels over time. Myjor x-axis ticks every 8 hours would fit the intervals better and also help the readers who usually think in days (i.e. 24 h not in 10/20/30 h etc.). Legend y-axis: % of initial fGCM concentration.

Reviewer #2: This is a very well written article. It is clearly explained and well executed. The methods for validation are sound and support the findings. The authors have effectively identified a new method that can used to noninvasively measure adrenal activity in pigs. I have no edits on the manuscript and would accept it in its current format. Great work.

6. PLOS authors have the option to publish the peer review history of their article (what does this mean?). If published, this will include your full peer review and any attached files.

Reviewer #1: Yes: Rupert Palme

Reviewer #2: No

---

## [Author Response · Author response to Decision Letter 0]

3 Feb 2020

Reviewers' comments:

Reviewer #1: The paper of Wolf et al (PONE-D-19-24903) aims at validating a method for evaluating adrenocortical activity via measuring faecal glucocorticoid metabolites (fGCMs) in pigs. The authors performed an ACTH challenge test (physiological validation) and a biological validation (transportation) in 6 female, immature pigs. They tested 4 different EIAs on a subset of sample (3 pigs) and found a 5α-pregnane-3β,11β,21-triol-20-one EIA best suited. Clear increases after ACTH injection and transportation in all 6 animals successfully validated the chosen EIA. An additional storage experiment demonstrated that FGCMs were stable up to 48 h. I think the experimental setup is well designed and the study adds knowledge about fGCMs analysis in an important species, where such methods have not been successfully validated so far. However, I miss a clear statement about the limitations/weaknesses of the study. The authors only used female, immature pigs. Because reproductive status (age) and sex are serious confounders in fGCM analyses, their influence needs to be elucidated (and the method validated in mature animals of both sexes) before a broad application of this method in pigs can be recommended.

We thank the reviewer for the valuable input and we hope that our revised version sufficiently addresses the respective points raised.

Detailed comments (ordered by appearance in the ms).

Title: I suggest indicating that only females (in female domestic pigs, or sows) were used (important as sex differences might be present). Better use “sample matrices” instead of “hormone matrices” (only metabolites are present in faeces)

The title has been adjusted as suggested 

Abstract: line 24: Replace “In order of not adding” with “To avoid”.

The sentence has been corrected (line 24)

Line28: “sample matrices” (also line 80, 150)

The sentence has been adjusted as suggested (lines 28, 77, 151)

I suggest to reword: “An assay measuring faecal glucocorticoid metabolites (fGCM) with a 3ß,11ß-diol group has proven suited to determine adrenocortical activity , showing…” (the antibody is not against those metabolites).

The sentence has been adjusted as suggested (line 28ff)

Line 31: Better: A cortisol EIA was used… (the immunogen is cortisol-3-CMO:BSA)

The sentence has been corrected (line 30ff)

Lines 33-38: That could be described shorter (especially as collection and transport from distant locations might be a problem in wildlife, but unlikely in domestic pigs, or?)

As suggested, the sentence has been shortened to assist the reader (although not common, longer transportation of domestic livestock occurs in SA) (line 35ff)

Line 52: stereotypic

The sentence has been corrected (line 52)

Line 67: Your focus is domestic livestock – so would not be another review better suited here?

Somewhere in the introduction, it might be worth mentioning that pigs are an especially stress susceptible species.

Alternative references have been provided. We also added a statement about the comparatively high stress susceptibility of pigs (line 47f) 

Line 85: ..on the stability of….

The sentence has been corrected (line 84f)

Line 90: I wonder when females of this breed get sexually mature, but 14 weeks old animals are too young.

In this breed puberty is reached between 115 and 160 days and they usually deliver their first set of piglets at about one year of age. 

Line 94: normally written in italics and with 2 words: ad libitum

The sentence has been corrected (line 94)

Line 103: 50 IU (10 µg) there is a space between a number and its dimension (many other cases too – e.g. line 134; 138, etc..).

We apologize for this oversight and added respective spaces between given numbers and their dimensions throughout the manuscript.

Line 104: Add country! Switzerland?

We added the country of origin (Switzerland)( line 105)

Line 106: Were samples from all defecations collected? 

Samples were taken as per scheduled timelines throughout the experiment as described in line 118ff. During the indicated period, samples from all voided faeces were collected (line 119).

Here and elsewhere: I suggest to replace the hyphen with a word space. 

Corrected 

Line 114: Salivette, or?

Yes, corrected (line 116)

Line 117-118: I guess the portion was taken from the core of each samples, and then mixed? Or why is it important to take it from the core of a well-mixed sample?

We apologize for the confusion. Yes, the portion was taken from the core of a sample, and then mixed. We adjusted the statement accordingly (line 120f)

Line 125: .. between defecation and freezing, or? For stability it is less important when the sample was collected!

Between defecation and freezing, we adjusted the text accordingly (line 127)

Line 128-129: I can imagine what was performed, but wording is unclear.

The sentence has been rephrased (line 130ff)

Line 142: via a cortisol enzyme immunoassay…. against cortisol-3-CMO:BSA.

The sentence has been corrected (line 144f)

Line 149: “Suited” is better than “reliable”. How would you define the latter? Most important is an EIA, which can monitor expected increases/decreases in adrenocortical activity well.

Corrected (line 151). 

Line 151 onwards: Four different EIAs were tested – please add EIA to each of the descriptions (e.g. 11-oxoaetiocholanolone I EIA) - I guess GB-spelling is chosen, so aetiocholanolone is correct.

The sentence has been corrected (line 154ff)

Lines 156-157: not possible to write by [number] – need to write the authors and then [].

The sentence has been corrected (line 158ff)

Line 159: delete “fGCM”

The sentence has been corrected (line 161)

Line 175-176: Peak fGCM concentrations following ACTH administration were expressed as percentage increase above baseline concentrations.

The sentence has been corrected (line 177f)

Line 183-186: How many samples per animal were present (I guess only baseline samples were used)?

The missing information was added (line 186ff). Only baseline samples were used for this part of the study.

Line 187: post defecation.

The sentence has been corrected (line 190)

Line 188: mean value … (no hormones in the faeces).

The sentence has been corrected (line 191)

Line 188/189: Unclear what the authors mean here? Why distribution?

The sentence has been rephrased (line 191ff)

Line 195: ..occuring

The wording has been corrected (line 198)

Line 209: enzyme immunoassays (EIAs)

The sentence has been corrected (line 212)

Table 1: The two 11-oxoaetiocholanolone have a higher increase – so why was the 5α-pregnane-3β,11β,21-triol-20-one EIA chosen? What means a “more stable” baseline? How is this made objective?

As stated in the manuscript, all four assays tested revealed an overall increase in fGCM concentrations above 100% post ACTH injection and thus appear suited for monitoring alterations in fGCM concentrations in sows. Although the two 11-oxoaetiocholanolone EIAs tested show a higher overall increase after ACTH administration compared to the 5α-pregnane-3β,11β,21-triol-20-one EIA, the pre-injection baseline as well as the post-injection peak values for the three animals tested were most comparable when using the 5α-pregnane-3β,11β,21-triol-20-one EIA. We further knew that with at least the 11-oxoaetiocholaonolone I EIA, fGCM concentrations were not stable after defecation (a 175% increase after 4 h, 375% after 24 h) (Möstl et al. 1999). That’s why we finally decided to continue with the 5α-pregnane-3β,11β,21-triol-20-one EIA. 

We adjusted the related statement which now reads ‘However, the 5α-pregnane-3β, 11β,21-triol-20-one EIA showed the most comparable baseline and peak values for all three animals and was chosen for the analysis of the remaining faecal samples.’ (line 207ff)

Line 213: (range: 215.7…) – but the 205.7 given in Table 1 is lower, why that?

Once the decision was made to continue with the 5α-pregnane-3β, 11β,21-triol-20-one EIA, the entire ACTH sample set (including the samples from the three animals used for EIA selection) were (re)analysed, which resulted in slight alterations in pre-injection baseline and post-injection peak values for the three animals already tested.

Fig 2 (and the same for Fig. 1): How were those 4 chosen (of course this makes someone curious about the other 2 ones (especially as Möstl et al., 1999 found even higher increases in some pigs, but no expressed ones in others). Please show all 6 pigs.

Only the individuals injected with Synacthen are shown in this figure, not the control individuals injected with saline. We added another figure depicting the two profiles of the control animals.

Line 232 (235) – what does “V” mean?

The V value is the sum of signed ranks (typically referred to as W). It has now been replaced by the corresponding Z value (lines 235, 238). 

Line 248: first time of “GIT” – spell it out!

Thanks for pointing that mistake out, we now refer to ‘gastro intestine transit (GIT) systems’ (line 251)

Line 249: add [8] to [14]

The reference has been added (line 252)

Line 250: “suited” is better than “sufficient”

The sentence has been corrected (line 253)

Line 251: [20] they also used a dexamethasone suppression test, and the assay was found unsuited, based (at least partly) on the fact that the suppression was not well reflected in fGCM levels.

The information has been added (line 254ff)

Line 250/251: Those studies did not perform ACTH test or biological validations “to measure fGCM concentrations” – please reword.

The sentence has been rephrased (line 254ff)

Line 256: .. a cortisol assay (using an antibody against cortisol-3-CMO:BSA) …

The sentences has been corrected (line 260f)

Line 259: barrows? Did you mean “farrows”?

Barrow can be used to describe castrated male pigs

Line 263: no hyphen in case of enzyme immunoassay. “glucocorticoid metabolites” – and also reword the rest (see comments above).

The sentence has been rephrased (line 269)

Line 265-267: Please reword. I wonder, but cannot find a sustained (up to 48 h) peak in Fig 2?

The sentence has been rephrased (line 270ff)

Line 268: baseline levels between 221.5 and 246.3 ng/g: I’m confused, as this is the third interval given for the baseline levels (Tab 1; line 213), and all are different?

We sincerely apologize for the confusion. As explained above, the different values stated in Tab1 compared to line 213 result from a re-measurement of the sub-sample set originally used for the assay screening. The values mentioned in line 268 are only referring to the animals treated with ACTH and not all six study animals (as referred to in line 213). To assist the reader we are now referring to all six animals here as well (line 272ff).

Line 269 – repetition of line 266 (180% increase)?

The sentence has been corrected (line 272ff).

Line 270: delete one full stop.

The mistake has been corrected (line 275).

Line 272: .. and the levels did not reach baseline after 24 hours [32].

The sentence has been corrected (line 277f).

Line 275: “fGCM recovery time”?

The sentence has been rephrased (line 279ff).

Line 276: .. and the appearance of their metabolites in the faeces [33].

The sentence has been corrected (line 281f).

Lines 283-286: I suggest adding “other species” and “different measurement method” as the most likely reasons for the observed differences.

The information has been added (line 289f).

LINE 288: “were” instead of “was”

The sentence has been corrected (line 294).

Line 294: .. when evaluating stressors in pigs using saliva as sample matrix.

The sentence has been corrected (line 299ff).

Line 296: ..might be a more suited..

The sentence has been corrected (line 302)

Line 299 “steroid concentrations”

The sentence has been corrected (line 305)

Lines 298-315: this is a rather long passage dealing with the stability experiment. I suggest focusing on domestic livestock and shortening it (e.g. the sentences lines 307-310 can be easily deleted without losing much). As mentioned before, I doubt there is much application in pigs for collection in remote locations and long sample transport.

The paragraph has been shortened (lines 304ff)

Line 322 onwards: I would prefer more straight forward applications in immature pigs (too many potential things/possibilities) – but see also further comments below!

In light of the aligned comments below we rephrased the conclusion section to address the various points raised

Lines 326-327: Don’t think that this statement is true. Your aim was validating such a non-invasive tool, but the importance must be beyond the tool.

The sentence has been removed

A clear statement about the limitations/weaknesses of the study needs to be added here. Only female, immature pigs were used. Thus results don’t guarantee a broad application of the method in pigs, because reproductive status (age) and sex are serious confounders in fGCM analyses, and thus their influence needs to be elucidated. That clearly limits the application of the method!

We agree with the reviewer and added a statement concerning the limitations (line 318ff). 

References: Not all words should be capitalized in the title [correct for 6, 9, 12,15, 17, 22, 28, 33, 36, 46].

The references have been corrected 

Line 337: last accessed?

The reference has been corrected (line 341).

Line 346: delete the 1 after cattle.

The reference has been corrected (line 351).

Line 379: Domestic Animal Endocrinology

The reference has been corrected (line 390).

Line 388: the journal is missing.

The reference has been corrected (line 399).

Line 393: “Suppl 2” is missing.

The reference has been corrected (line 405).

Line 395: Veterinary Research Communications

The reference has been corrected (line 408).

Line 452: “ethological”

The reference has been corrected (line 457).

Line 453: better: “last accessed” than “updated”.

The reference has been corrected (line 459)

Fig 1 and 2: Smaller symbols for the 4 pigs would make the picture clearer, also allowing adding the missing two.

We have changed both figures accordingly and added additional figures with the control animals.

Fig. 2. What about the animal which showed a peak in fGCM concentrations at the time of ACTH injection? Was there a known stressor occurring the day before? There seemed to be a smaller peak (ACTH related) present on Day 1 (based on the broader shoulder).

Unfortunately we don’t have any data on this point and can only speculate about an additional stressor perceived on that day. 

Fig. 3: The y-axis should be scaled from 0 to 100. This is more correct and also better representing the message of only insignificant changes in FGCM levels over time. Myjor x-axis ticks every 8 hours would fit the intervals better and also help the readers who usually think in days (i.e. 24 h not in 10/20/30 h etc.). Legend y-axis: % of initial fGCM concentration.

We have changed figure 3 accordingly

Reviewer #2: This is a very well written article. It is clearly explained and well executed. The methods for validation are sound and support the findings. The authors have effectively identified a new method that can used to noninvasively measure adrenal activity in pigs. I have no edits on the manuscript and would accept it in its current format. Great work.

We thank the reviewer for the encouraging words.

---

## [Decision Letter · Decision Letter 1]

20 Apr 2020

PONE-D-19-24903R1

Non-invasive monitoring of adrenocortical function in female domestic pigs using saliva and faeces as sample matrices

PLOS ONE

Dear Dr Wolf,

Thank you for submitting your manuscript to PLOS ONE. After careful consideration, we feel that it has merit but does not fully meet PLOS ONE’s publication criteria as it currently stands. Therefore, we invite you to submit a revised version of the manuscript that addresses the points raised during the review process.

- fix the methods descriptions

- compare saliva and faeces data

- fix minor issues with legends etc.

We would appreciate receiving your revised manuscript by Jun 04 2020 11:59PM. To enhance the reproducibility of your results, we recommend that if applicable you deposit your laboratory protocols in protocols.io, where a protocol can be assigned its own identifier (DOI) such that it can be cited independently in the future. For instructions see: http://journals.plos.org/plosone/s/submission-guidelines#loc-laboratory-protocols

We look forward to receiving your revised manuscript.

Kind regards,

Henrik Oster, Ph.D.

Academic Editor

PLOS ONE

Reviewers' comments:

Reviewer's Responses to Questions

**Comments to the Author**

1. If the authors have adequately addressed your comments raised in a previous round of review and you feel that this manuscript is now acceptable for publication, you may indicate that here to bypass the “Comments to the Author” section, enter your conflict of interest statement in the “Confidential to Editor” section, and submit your "Accept" recommendation.

Reviewer #1: All comments have been addressed

Reviewer #3: All comments have been addressed

2. Is the manuscript technically sound, and do the data support the conclusions?

Reviewer #1: Yes

Reviewer #3: Partly

3. Has the statistical analysis been performed appropriately and rigorously? 

Reviewer #1: Yes

Reviewer #3: I Don't Know

4. Have the authors made all data underlying the findings in their manuscript fully available?

Reviewer #1: No

Reviewer #3: Yes

5. Is the manuscript presented in an intelligible fashion and written in standard English?

Reviewer #1: Yes

Reviewer #3: Yes

6. Review Comments to the Author

Reviewer #1: Thank you – the revised paper is significantly improved now. The authors have sufficiently addressed my points; thus, I think the paper is suitable for publication now. Only a few minor errors are left for correction (typos; etc… - see details below), before it should go to the production department.

The hyphen of “enzyme-immunoassay” was only partly deleted – so there is now an inconsistent situation. Please also delete it in the remaining cases (lines 144 and 153)

The 11-oxoaetiocholanolone is sometimes still present in its US-spelling (please change “e” to “ae”; lines 154 and 155)

Thanks for adding data of the control animals (Fig. 2b). I’ve only one question left now here: The mentioned increase after saline injection (60% above baseline; lines 219/220) is present on Day 0 – when were the samples collected in relation to the injection time – I wonder if that really reflects the saline injection? If too close, I suggest deleting that sentence.

Line 260: please delete the first “”an assay using”

The problem with the capitalized title words is not fully resolved – please change all occurrences in refs 6.; 8; 10; 18 and 35 (correct only with the first word after a “:”).

Line 401: A typing error was introduced: “acpacity”.

Ref 33: “American”

Ref 34: The end was lost during revision; please add: Wiener Tierärztliche Monatsschrift. 2010;97: 259–262.

Reviewer #3: The revised paper is well-written and the goals of the study are clear and well-justified. The results of the studies described will be of use to others that would like to implement non-invasive fecal glucocorticoid metabolite measures with domestic pigs. The methods appear sound, although there are some details that are missing or presented in a confusing manner that makes it difficult to interpret the significance of the results. There are important changes to address to help readers interpret the study design and results, however these should be relatively easy to address.

Introduction:

The background information is clearly presented and the motivation for the current study is clear. The study objectives are sound.

Methods:

Overall, the methods are well-explained and easy to follow. The different sub-studies are well-designed, and the sampling regime was good.

There are three sub-studies in the current work: (1) document post-ACTH and post transport salivary/fecal corticoid peak concentrations and timing, (2) compare morning vs. evening salivary/fecal corticoid concentrations, and (3) determine relative degradation of glucocorticoid metabolites in feces kept in ambient conditions. These three goals/sub-studies are not presented as distinct, separate studies in the methods and results, and this makes it difficult at times to elucidate the sample size and sampling/analytical methods associated with each result. One way to clarify the methods would be to provide salivary and fecal sampling times relative to each manipulation (e.g. ACTH, transport, time of day). This is well done for the degradation study, but difficult to understand for the ACTH/transport studies. Please also see comments about Results below.

One example of this confusion: there is disagreement in the description of when fecal samples were collected in the methods vs. how data are presented in Figure 2. In the Methods, the authors indicated that fecal samples were collected during the same 7.5-hour continuous window on a daily basis (8:00-15:30 h). However, in Figure 2, there are data points that indicate that samples were collected every 12 hours rather than every 24 hours (i.e. the data points after ACTH injection). This should be clarified in both the Methods and the Results. For example, the authors should indicate when ACTH injections were administered relative to fecal sample collections.

In the above example, there is another concern because a peak in fGCM is more readily documented if samples are collected more frequently (as they appear to have been done post- vs. pre-ACTH injection). Specifically, when samples are collected more frequently, there is more chance to catch a true peak without averaging a peak sample with other lower-concentration samples collected during a wider sampling window. Thus, if the sampling regime became more frequent after the ACTH injection compared to before the injection, then there is some concern about methodological bias leading to a greater chance of documenting peak fGCM after vs. before ACTH injection. The authors should clarify this.

It is good to see that the authors compared multiple EIAs.

To compare morning vs. afternoon salivary cortisol and fGCM it would be more appropriate to compare corticoid concentrations from samples that are collected at more distinct time points. In the present design, the authors compare ‘morning’ to ‘afternoon’ corticoid concentrations by comparing corticoid concentrations from samples collected between 8:00-12:00 h (morning) vs. 12:01-18:00 h (afternoon). These are not particularly distinct time frames – the morning samples may consist of a majority of samples collected near 11:00 h while afternoon samples could be primarily from 12:30 h. To truly determine if there is a circadian rhythm, it would be best to compare samples that are collected at the beginning vs. the end of the active period. To truly analyze if there is a circadian rhythm in fGCM, the authors should compare mean fGCM from 8:00-12:00 h vs. 14:00-18:00 h, or better yet, compare fGCM from samples collected 8:00-10:00 vs. 16:00-18:00 h. As currently conducted, this analysis favors the authors hypothesis that fGCM measures will fluctuate less across the day compared to salivary cortisol measures.

Were all animals equally represented in the morning vs. afternoon samples? This should be clarified.

The authors should indicate the sample size involved in each statistical analysis. And to indicate how often each animal was represented in each data set (e.g. was animal ID controlled for in analyses that included multiple samples from the same individual?).

In the current study, blood samples were not collected to allow for a comparison of fGCM concentrations to blood cortisol concentrations as a validation step. However, both saliva and feces were collected from the same six pigs. Can the authors include an analysis to compare individual salivary and fecal corticoid measures? In other words, were pigs that had high baseline or post-ACTH salivary cortisol levels the same pigs that had high fGCM concentrations at these times? This is not necessary, but it would provide additional validation of the fGCM method used in the current study.

Results:

Please include sample size in all reported result that included an average or median value (in particular for ‘Effect of Collection Time’ and ‘Stability test for fGCM concentrations’ results).

Line 216: Clarify that the results in this section are from the 5α-pregnane-3β,11β,21-triol-20-one EIA. In addition, explain how the average baseline fGCM level was calculated. How many samples were included from each female? Were individual female means calculated on each day or across multiple days, and then an average taken from all individual female averages? Please indicate how differential sampling from individual females was accounted for in the calculations, or at least provide information on how much each female contributed to the sample size.

Figure 2 legend: Indicate that values in figure are from the 5α-pregnane-3β,11β,21-triol-20-one EIA.

Line 241: “Within the first hour,” should read “Within the first hour at ambient temperature,”

Discussion:

The authors reference two papers that previously used fecal corticoid measures in domestic pigs with ‘mixed results’. Given the relative importance of these prior papers relative to the current manuscript, the authors should tell us more about why the methods used in the current study led to clear results, while the prior work led to mixed results. What were the mixed results in prior studies? What methods did they use? And why were those prior studied unsuccessful while the current study was successful? These are important aspects that should be clarified.

Otherwise, the Discussion appropriate compares results of the current study to results from other relevant studies.

7. PLOS authors have the option to publish the peer review history of their article (what does this mean?). If published, this will include your full peer review and any attached files.

Reviewer #1: Yes: Rupert Palme

Reviewer #3: No

---

## [Author Response · Author response to Decision Letter 1]

12 May 2020

Response to editor and reviewers

PONE-D-19-24903R1

Non-invasive monitoring of adrenocortical function in female domestic pigs using saliva and faeces as sample matrices

Dear Dr Wolf,

Thank you for submitting your manuscript to PLOS ONE. After careful consideration, we feel that it has merit but does not fully meet PLOS ONE’s publication criteria as it currently stands. Therefore, we invite you to submit a revised version of the manuscript that addresses the points raised during the review process.

- fix the methods descriptions

- compare saliva and faeces data

- fix minor issues with legends etc.

We look forward to receiving your revised manuscript.

Kind regards,

Henrik Oster, Ph.D.

Academic Editor

PLOS ONE

We thank the editor for the opportunity to resubmit our revised manuscript. As suggested, we have addressed the comments from the reviewers and edited the manuscript accordingly if required. 

 

Review Comments to the Author

Reviewer #1: Thank you – the revised paper is significantly improved now. The authors have sufficiently addressed my points; thus, I think the paper is suitable for publication now. Only a few minor errors are left for correction (typos; etc… - see details below), before it should go to the production department.

We thank the reviewer and hope we have addressed the remaining points raised. 

The hyphen of “enzyme-immunoassay” was only partly deleted – so there is now an inconsistent situation. Please also delete it in the remaining cases (lines 144 and 153)

We have now corrected the spelling (lines 153 & 162)

The 11-oxoaetiocholanolone is sometimes still present in its US-spelling (please change “e” to “ae”; lines 154 and 155)

We have now corrected the spelling (lines 163 & 164)

Thanks for adding data of the control animals (Fig. 2b). I’ve only one question left now here: The mentioned increase after saline injection (60% above baseline; lines 219/220) is present on Day 0 – when were the samples collected in relation to the injection time – I wonder if that really reflects the saline injection? If too close, I suggest deleting that sentence.

We agree with the reviewer and omitted the statement

Line 260: please delete the first “”an assay using”

We have corrected the sentence (line 277)

The problem with the capitalized title words is not fully resolved – please change all occurrences in refs 6.; 8; 10; 18 and 35 (correct only with the first word after a “:”).

Line 401: A typing error was introduced: “acpacity”.

Ref 33: “American”

Ref 34: The end was lost during revision; please add: Wiener Tierärztliche Monatsschrift. 2010;97: 259–262.

We have now corrected the references

 

Reviewer #3: The revised paper is well-written and the goals of the study are clear and well-justified. The results of the studies described will be of use to others that would like to implement non-invasive fecal glucocorticoid metabolite measures with domestic pigs. The methods appear sound, although there are some details that are missing or presented in a confusing manner that makes it difficult to interpret the significance of the results. There are important changes to address to help readers interpret the study design and results, however these should be relatively easy to address.

We thank the reviewer for the valuable input and hope that our revised version sufficiently addresses the respective points raised.

Introduction:

The background information is clearly presented and the motivation for the current study is clear. The study objectives are sound.

Methods:

Overall, the methods are well-explained and easy to follow. The different sub-studies are well-designed, and the sampling regime was good.

There are three sub-studies in the current work: (1) document post-ACTH and post transport salivary/fecal corticoid peak concentrations and timing, (2) compare morning vs. evening salivary/fecal corticoid concentrations, and (3) determine relative degradation of glucocorticoid metabolites in feces kept in ambient conditions. These three goals/sub-studies are not presented as distinct, separate studies in the methods and results, and this makes it difficult at times to elucidate the sample size and sampling/analytical methods associated with each result. One way to clarify the methods would be to provide salivary and fecal sampling times relative to each manipulation (e.g. ACTH, transport, time of day). This is well done for the degradation study, but difficult to understand for the ACTH/transport studies. Please also see comments about Results below.

To assist the reader, we have now added more information concerning the timeline of events and sampling regime, and subdivided the methods section further into the specific parts of the study, namely the monitoring of transport and the effect of sampling time.

One example of this confusion: there is disagreement in the description of when fecal samples were collected in the methods vs. how data are presented in Figure 2. In the Methods, the authors indicated that fecal samples were collected during the same 7.5-hour continuous window on a daily basis (8:00-15:30 h). However, in Figure 2, there are data points that indicate that samples were collected every 12 hours rather than every 24 hours (i.e. the data points after ACTH injection). This should be clarified in both the Methods and the Results. For example, the authors should indicate when ACTH injections were administered relative to fecal sample collections. 

We apologise for the confusion and have now added a more detailed description of the sampling in the methods section. We now also indicate that ACTH administration took place in the early morning of day 0 (line 103). We also want to point out that Figure 2 shows the median fGCM levels per day for each individual, and only on the 2 days after injection we show fGCM values of all individual samples to better visualize the time when the peak concentrations were found. 

In the above example, there is another concern because a peak in fGCM is more readily documented if samples are collected more frequently (as they appear to have been done post- vs. pre-ACTH injection). Specifically, when samples are collected more frequently, there is more chance to catch a true peak without averaging a peak sample with other lower-concentration samples collected during a wider sampling window. Thus, if the sampling regime became more frequent after the ACTH injection compared to before the injection, then there is some concern about methodological bias leading to a greater chance of documenting peak fGCM after vs. before ACTH injection. The authors should clarify this.

We apologise for the misunderstanding, the sampling regime over the course of the ACTH challenge did not change, as samples were taken from all faeces voided between 8:00 and 18:00h. We now indicate that in the revised version of the manuscript (line 123). This should indicate, that there is no bias between the samples collected before and after ACTH injection.

It is good to see that the authors compared multiple EIAs.

To compare morning vs. afternoon salivary cortisol and fGCM it would be more appropriate to compare corticoid concentrations from samples that are collected at more distinct time points. In the present design, the authors compare ‘morning’ to ‘afternoon’ corticoid concentrations by comparing corticoid concentrations from samples collected between 8:00-12:00 h (morning) vs. 12:01-18:00 h (afternoon). These are not particularly distinct time frames – the morning samples may consist of a majority of samples collected near 11:00 h while afternoon samples could be primarily from 12:30 h. To truly determine if there is a circadian rhythm, it would be best to compare samples that are collected at the beginning vs. the end of the active period. To truly analyze if there is a circadian rhythm in fGCM, the authors should compare mean fGCM from 8:00-12:00 h vs. 14:00-18:00 h, or better yet, compare fGCM from samples collected 8:00-10:00 vs. 16:00-18:00 h. As currently conducted, this analysis favors the authors hypothesis that fGCM measures will fluctuate less across the day compared to salivary cortisol measures.

We agree with the reviewer that this is a valuable point, originally, we used fGCM values from samples for the ‘afternoon’ sample set collected between 12:45 and 18:00 for this analysis. However, when excluding samples collected before 16:00, the mean of the afternoon samples appears comparable to the mean including the early afternoon samples (235.6 vs 240.8 ng/g DW; W=5134, p=0.738). All morning samples were collected between 8:00 and 10:00am, and fGCM concentrations did not differ significantly regardless of comparing steroid values from the morning samples with those from the more broader (12:45 – 18:00) or stricter (16:00 – 18:00) afternoon sample set. We therefore decided to keep the currently presented approach. However, we are willing to exchange it with the one representing the more rigorous time bouts if requested.

Were all animals equally represented in the morning vs. afternoon samples? This should be clarified.

The authors should indicate the sample size involved in each statistical analysis. And to indicate how often each animal was represented in each data set (e.g. was animal ID controlled for in analyses that included multiple samples from the same individual?).

All individuals are represented equally in morning and afternoon samples as indicated in the methods section (line 132ff & 194ff). 6 morning samples were compared to 6 afternoon samples from the respective days. It is therefore not necessary to control for individual bias within the analyses. 

In the current study, blood samples were not collected to allow for a comparison of fGCM concentrations to blood cortisol concentrations as a validation step. However, both saliva and feces were collected from the same six pigs. Can the authors include an analysis to compare individual salivary and fecal corticoid measures? In other words, were pigs that had high baseline or post-ACTH salivary cortisol levels the same pigs that had high fGCM concentrations at these times? This is not necessary, but it would provide additional validation of the fGCM method used in the current study.

Salivary GC concentrations, similar to GC concentrations in blood, provide a valuable insight into GC fluctuations over a small timeframe by indicating what is going on in the organism seconds to minutes before the sample was collected (Anestis 2010). They are therefore also prone to additional factors like diurnal rhythm of GC production. Faecal samples on the other hand provide a more cumulative picture of GCM concentrations, providing an insight into a time depth of up to 40 hours, depending on the size of the organism (Touma et al. 2004; Anestis 2010; Behringer & Deschner 2017). As animals, even if kept under standardised conditions, might perceive various occurring stressors differently, the cumulative signal depicted in fGCM concentrations will most likely differ from that point-in-time GC value revealed in a saliva sample. We therefore don’t think that a comparison of salivary GC and fGCM concentrations would be reasonable and thus haven’t included any additional information in the revised version of the manuscript. However, we did a respective comparison of individual baseline sGC/fGCM levels, and as expected the animals with the highest or lowest sGC concentrations do not have the highest or lowest fGCM levels. Interestingly, and another possible reason for the mismatch might be that especially the fGCM baseline values only vary minimally between the individuals (216 – 246 ng/g DW) and could thus be considered equal. 

Results:

Please include sample size in all reported result that included an average or median value (in particular for ‘Effect of Collection Time’ and ‘Stability test for fGCM concentrations’ results).

The sample size per individual for the analyses of the collection time is indicated in the methods, we now added the total numbers (line 198). The sample size for the stability test is 24 as already indicated in the methods (line 138). 

Line 216: Clarify that the results in this section are from the 5α-pregnane-3β,11β,21-triol-20-one EIA. In addition, explain how the average baseline fGCM level was calculated. How many samples were included from each female? Were individual female means calculated on each day or across multiple days, and then an average taken from all individual female averages? Please indicate how differential sampling from individual females was accounted for in the calculations, or at least provide information on how much each female contributed to the sample size.

The EIA name has been added to the sentence. ‘To establish a baseline individual median fGCM concentrations were calculated using pre- ACTH administration samples’ (line 185). 9 samples per individual (collected 5 days prior to ACTH injection) were used to establish individual baseline levels. We have added more information (line 99ff). 

Figure 2 legend: Indicate that values in figure are from the 5α-pregnane-3β,11β,21-triol-20-one EIA.

The information has been added to the figure legend (line 234)

Line 241: “Within the first hour,” should read “Within the first hour at ambient temperature,”

The sentence has been corrected (line 255)

Discussion:

The authors reference two papers that previously used fecal corticoid measures in domestic pigs with ‘mixed results’. Given the relative importance of these prior papers relative to the current manuscript, the authors should tell us more about why the methods used in the current study led to clear results, while the prior work led to mixed results. What were the mixed results in prior studies? What methods did they use? And why were those prior studied unsuccessful while the current study was successful? These are important aspects that should be clarified.

Additional information concerning the used methods and outcome in these studies has been added to the discussion (line 270ff)

Otherwise, the Discussion appropriate compares results of the current study to results from other relevant studies.

---

## [Decision Letter · Decision Letter 2]

8 Jun 2020

Non-invasive monitoring of adrenocortical function in female domestic pigs using saliva and faeces as sample matrices

PONE-D-19-24903R2

Dear Dr. Wolf,

We’re pleased to inform you that your manuscript has been judged scientifically suitable for publication and will be formally accepted for publication once it meets all outstanding technical requirements.

Kind regards,

Henrik Oster, Ph.D.

Academic Editor

PLOS ONE

Additional Editor Comments (optional):

Reviewers' comments:

Reviewer's Responses to Questions

**Comments to the Author**

1. If the authors have adequately addressed your comments raised in a previous round of review and you feel that this manuscript is now acceptable for publication, you may indicate that here to bypass the “Comments to the Author” section, enter your conflict of interest statement in the “Confidential to Editor” section, and submit your "Accept" recommendation.

Reviewer #3: (No Response)

2. Is the manuscript technically sound, and do the data support the conclusions?

Reviewer #3: Yes

3. Has the statistical analysis been performed appropriately and rigorously? 

Reviewer #3: Yes

4. Have the authors made all data underlying the findings in their manuscript fully available?

Reviewer #3: Yes

5. Is the manuscript presented in an intelligible fashion and written in standard English?

Reviewer #3: Yes

6. Review Comments to the Author

Reviewer #3: The authors have addressed the majority of my comments and concerns on the original revised manuscript.

The description of the methods and the number of samples involved in each analysis has improved.

I appreciate the authors’ legitimate concerns with comparing salivary and fecal corticoid levels. Unfortunately, they do not take full advantage of the data that they have; there are ways to control for salivary corticoid moment-to-moment variation to legitimize comparison with more integrated fecal measures. For example, salivary levels could be adjusted for time of day, or an integrated salivary corticoid metric (e.g. AUC) post-ACTH injection could be compared to post-ACTH fecal measures, etc. I understand that the data set is small, and individual variability in corticoid production may be relatively limited given the housing conditions, thus limiting the possibility of finding statistically significant results.

For the time of day analysis of fecal corticoids, I would include the more stringent time of day analysis.

Minor:

- There are 5 (not 6) days of fecal sample collections prior to the ACTH injection day.

- Page 6, line 131: It would help to indicate how many saliva samples were collected pre- vs. post-ACTH injection – as was done for fecal samples.

7. PLOS authors have the option to publish the peer review history of their article (what does this mean?). If published, this will include your full peer review and any attached files.

Reviewer #3: Yes: Sonia A. Cavigelli

---

## [Editor Report · Acceptance letter]

12 Jun 2020

PONE-D-19-24903R2 

Non-invasive monitoring of adrenocortical function in female domestic pigs using saliva and faeces as sample matrices 

Dear Dr. Wolf:

I'm pleased to inform you that your manuscript has been deemed suitable for publication in PLOS ONE. Congratulations! Your manuscript is now with our production department. 

Kind regards, 

on behalf of

Prof. Henrik Oster 

Academic Editor

PLOS ONE